# Cross-language sentiment analysis of European Twitter messages during the COVID-19 pandemic

**Anna Kruspe**
German Aerospace Center (DLR)
Institute of Data Science
Jena, Germany
`anna.kruspe@dlr.de`

**Matthias Häberle**
Technical University of Munich (TUM)
Signal Processing in Earth Observation (SiPEO)
Munich, Germany
`matthias.haeberle@tum.de`

**Iona Kuhn**
German Aerospace Center (DLR)
Institute of Data Science
Jena, Germany
`iona.kuhn@dlr.de`

**Xiao Xiang Zhu**
German Aerospace Center (DLR)
Remote Sensing Technology Institute (IMF)
Oberpfaffenhofen, Germany
`xiaoxiang.zhu@dlr.de`

## Abstract

Social media data can be a very salient source of information during crises. User-generated messages provide a window into people's minds during such times, allowing us insights about their moods and opinions. Due to the vast amounts of such messages, a large-scale analysis of population-wide developments becomes possible.

In this paper, we analyze Twitter messages (tweets) collected during the first months of the COVID-19 pandemic in Europe with regard to their sentiment. This is implemented with a neural network for sentiment analysis using multilingual sentence embeddings. We separate the results by country of origin, and correlate their temporal development with events in those countries. This allows us to study the effect of the situation on people's moods. We see, for example, that lockdown announcements correlate with a deterioration of mood in almost all surveyed countries, which recovers within a short time span.

## 1 Introduction

The COVID-19 pandemic has led to a worldwide situation with a large number of unknowns. Many heretofore unseen events occurred within a short time span, and governments have had to make quick decisions for containing the spread of the disease. Due to the extreme novelty of the situation, the outcomes of many of these events have not been studied well so far. This is true with regards to their medical effect, as well as the effect on people's perceptions and moods.

First studies about the effect the pandemic has on people's lives are being published at the moment (e.g. Betsch et al., 2020), mainly focusing on surveys and polls. Naturally, such studies are limited to relatively small numbers of participants and focus on specific regions (e.g. countries).

In contrast, social media provides a large amount of user-created messages reflective of those users' moods and opinions. The issue with this data source is the difficulty of analysis - social media messages are extremely noisy and idiosyncratic, and the amount of incoming data is much too large to analyze manually. We therefore need automatic methods to extract meaningful insights.

In this paper, we describe a data set collected from Twitter during the months of December 2019 through April 2020, and present an automatic method for determining the sentiments contained in these messages. We then calculate the development of these sentiments over time, segment the results by country, and correlate them with events that took place in each country during those five months.

## 2 Related work

Since the pandemic outbreak and lockdown measures, numerous studies have been published to investigate the impact of the corona pandemic on

Twitter.

Feng and Zhou (2020) analyzed tweets from the US on a state and county level. First, they could detect differences in temporal tweeting patterns and found that people tweeting more about COVID-19 during working hours as the pandemic progressed. Furthermore, they conducted a sentiment analysis over time including an event specific subtask reporting negative sentiment when the 1000th death was announced and positive when the lockdown measures were eased in the states.

Lyu et al. (2020) looked into US-tweets which contained the terms "Chinese-virus" or "Wuhan-virus" referring to the COVID-19 pandemic to perform a user characterization. They compared the results to users that did not make use of such controversial vocabulary. The findings suggest that there are noticeable differences in age group, geo-location, or followed politicians.

Chen et al. (2020) focused on sentiment analysis and topic modelling on COVID-19 tweets containing the term "Chinese-virus" (controversial) and contrasted them against tweets without such terms (non-controversial). Tweets containing "Chinese-virus" discussing more topics which are related to China whereas tweets without such words stressing how to defend the virus. The sentiment analysis revealed for both groups negative sentiment, yet with a slightly more positive and analytical tone for the non-controversial tweets. Furthermore, they accent more the future and what the group itself can do to fight the disease. In contrast, the controversial group aiming more on the past and concentrate on what others should do.

## 3 Data collection

For our study, we used the freely available Twitter API to collect the tweets from December 2019 to April 2020. The free API allows streaming of 1% of the total tweet amount. To cover the largest possible area, we used a bounding box which includes the entire world. From this data, we sub-sampled 4,683,226 geo-referenced tweets in 60 languages located in the Europe. To create the Europe sample, we downloaded a shapefile of the earth[1], then we filtered by country performing a point in polygon test using the Python package *Shapely*[2]. Figure 1 depicts the Europe Twitter activity in total numbers.

Most tweets come from the U.K. Tweets are not filtered by topic, i.e. many of them are going to be about other topics than COVID-19. This is by design. As we will describe later, we also apply a simple keyword filter to detect tweets that are probably COVID-19-related for further analysis.

## 4 Analysis method

We now describe how the automatic sentiment analysis was performed, and the considerations involved in this method.

### 4.1 Sentiment modeling

In order to analyze these large amounts of data, we focus on an automatic method for sentiment analysis. We train a neural network for sentiment analysis on tweets. The text input layer of the network is followed by a pre-trained word or sentence embedding. The resulting embedding vectors are fed into a 128-dimensional fully-connected ReLU layer with 50% dropout, followed by a regression output layer with sigmoid activation. Mean squared error is used as loss. The model is visualized in figure 2.

This network is trained on the *Sentiment140* dataset (Go et al., 2009). This dataset contains around 1.5 million tweets collected through keyword search, and then annotated automatically by detecting emoticons. Tweets are determined to have positive, neutral, or negative sentiment. We map these sentiments to the values 1.0, 0.5, and 0.0 for the regression. Sentiment for unseen tweets is then represented on a continuous scale at the output.

We test variants of the model using the following pre-trained word- and sentence-level embeddings:

- A skip-gram version of *word2vec* (Mikolov et al., 2013) trained on the English-language Wikipedia[3]

- A multilingual version of BERT (Devlin et al., 2018) trained on Wikipedia data[4]

- A multilingual version of BERT trained on 160 million tweets containing COVID-19 keywords[5] (Müller et al., 2020)

---

[1]https://www.naturalearthdata.com/downloads/10m-cultural-vectors/10m-admin-0-countries/
[2]https://pypi.org/project/Shapely/

[3]https://tfhub.dev/google/Wiki-words-250/2
[4]https://tfhub.dev/tensorflow/bert_multi_cased_L-12_H-768_A-12/2
[5]https://tfhub.dev/digitalepidemiologylab/covid-twitter-bert/1

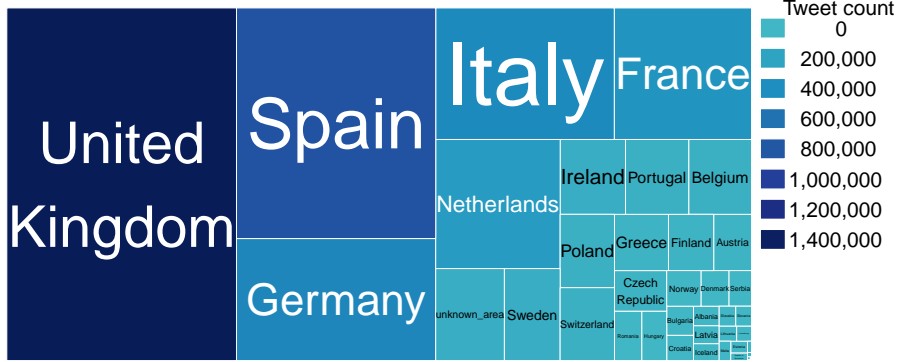

Figure 1: Treemap of Twitter activity in Europe during the time period of December 2019 to April 2020.

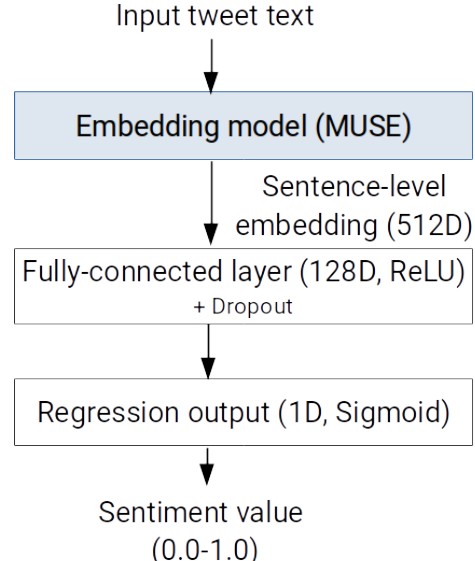

Figure 2: Architecture of the sentiment analysis model.

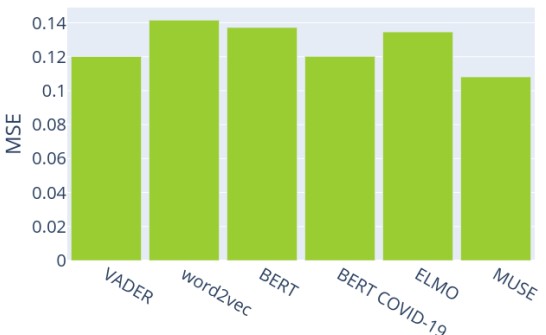

Figure 3: MSE for different models on the *Sentiment140* test dataset.

• An ELMO model (Peters et al., 2018) trained on the 1 Billion Word Benchmark dataset[6]

• The Multilingual Universal Sentence Encoder (MUSE)[7] (Yang et al., 2019)

We train each sentiment analysis model on the *Sentiment140* dataset for 10 epochs. Mean squared error results on the unseen test portion of the same dataset are shown in figure 3. For comparison, we also include an analysis conducted by VADER which is a rule-based sentiment reasoner designed for social media messages (Hutto and Gilbert, 2014).

Interestingly, most neural network results are in the range of the rule-based approach. BERT delivers better results than the *word2vec* model, with ELMO and the COVID-19-specific version also leading to improvements. However, the best result is achieved with the pre-trained multilingual USE model, which can embed whole sentences rather than (contextualized) words. We therefore perform the subsequent sentiment analysis with the MUSE-based model.

An interesting side note here is that the dataset only contains English-language tweets, but the sentence embedding is multilingual (for 16 languages). We freeze the embedding weights to prevent them from over-adapting to English. Due to the cross-lingual semantic representation capabilities of the pre-trained embedding, we expect the model to be able to detect sentiment in other languages just as well.

With the created model, we perform sentiment analysis on the 4.6 million tweets collected

---

[6]https://tfhub.dev/google/elmo/3
[7]https://tfhub.dev/google/universal-sentence-encoder-multilingual/3

from December to April, and then aggregate the results over time. This provides us with a representation of the development of Twitter messages' average sentiment over time. We specifically consider all collected tweets rather than just those determined to be topically related to COVID-19 because we are interested in the effect on people's moods in general, not just with regards to the pandemic. Additionally, we also filter the tweets by COVID-19-associated keywords, and analyze their sentiments as well. The chosen keywords are listed in figure 4.

| | | |
|---|---|---|
| corona | コロナ | कोरोना |
| covid | 冠狀病毒 | โคโรน่า |
| wuhan | 武漢 | करोना |
| koroona | ਕੋਰੋਨਾ | קורונה |
| корона | 코로나 | کورونا |
| κορωνα | կորոնա | ویروس |
| korona | | |

Figure 4: Keywords used for filtering the tweets (not case sensitive).

## 4.2 Considerations

There are some assumptions implicit in this analysis method that we want to address here. First of all, we only consider tweets containing a geolocation. This applies to less than 1% of the whole tweet stream, but according to Sloan et al. (2013), the amount of geolocated tweets closely follows the geographic population distribution. According to Graham et al. (2014), there probably are factors determining which users share their locations and which ones do not, but there is no systematic study of these.

Other assumptions arise from the analysis method itself. For one, we assume that the model is able to extract meaningful sentiment values from the data. However, sentiment is subjective, and the model may be failing for certain constructs (e.g. negations, sarcasm). Additionally, modeling sentiment on a binary scale does not tell the whole story. "Positive" sentiment encompasses, for example, happy or hopeful tweets, "negative" angry or sad tweets, and "neutral" tweets can be news tweets, for example. A more finegrained analysis would be of interest in the future.

We also assume a somewhat similar perception of sentiment across languages. Finally, we assume that the detected sentiments as a whole are reflective of the mood within the community; on the other hand, mood is not quantifiable in the first place. All of these assumptions can be called into question. Nevertheless, while they may not be applicable for every single tweet, we hope to detect interesting effects on a large scale. When analyzing thousands of tweets within each time frame, random fluctuations become less likely. We believe that this analysis can provide useful insights into people's thoughts, and form an interesting basis for future studies from psychological or sociological perspectives.

## 5 Results

In the following, we present the detected sentiment developments over time over-all and for select countries, and correlate them with events that took place within these months. Results for some other countries would have been interesting as well, but were not included because the main spoken language is not covered by MUSE (e.g. Sweden, Denmark). Others were excluded because there was not enough material available; we only analyze countries with at least 300,000 recorded tweets. As described in section 3, tweets are filtered geographically, not by language (i.e. Italian tweets may also be in other languages than Italian).

### 5.1 Over-all

In total, we analyzed around 4.6 million tweets, of which around 79,000 contained at least one COVID-19 keyword. Figure 5 shows the development of the sentiment over time for all tweets and for those with keywords, as well as the development of the number of keyworded tweets. The sentiment results are smoothed on a weekly basis (otherwise, we would be seeing a lot of movement during the week, e.g. an increase on the weekends). For the average over all tweets, we see a slight decrease in sentiment over time, indicating possibly that users' moods deteriorated over the last few months. There are some side effects that need to be considered here. For example, the curve rises slightly for holidays like Christmas and Easter (April 12). Interestingly, we see a clear dip around mid-March. Most European countries started implementing strong social distancing measures around this time. We will talk about this in more detail in the next sections.

We see that keywords were used very rarely before mid-January, and only saw a massive increase in

usage around the beginning of March. Lately, usage has been decreasing again, indicating a loss of interest over time. Consequently, the sentiment analysis for keyword tweets is not expressive in the beginning. Starting with the more frequent usage in February, the associated sentiment drops massively, indicating that these tweets are now used in relation with the pandemic. Interestingly, the sentiment recovers with the increased use in March - it is possible that users were starting to think about the risks and handling of the situation in a more relaxed way over time. Still, the sentiment curve for keyword tweets lies significantly below the average one, which is to be expected for this all-around rather negative topic.

## 5.2 Analysis by country

We next aggregated the tweets by country as described in section 3 and performed the same analysis by country. The country-wise curves are shown jointly in figure 6. Comparing the absolute average sentiment values between countries is difficult as they may be influenced by the languages or cultural factors. However, the relative development is interesting. We see that all curves progress in a relatively similar fashion, with peaks around Christmas and Easter, a strong dip in the middle of March, and a general slow decrease in sentiment. In the following, we will have a closer look at each country's development. (Note that the keyword-only curves are cut of in the beginning for some countries due to a low number of keyword tweets).

### 5.2.1 Italy

Figure 7 shows the average sentiment for all Italian tweets and all Italian keyword tweets, as well as the development of keyword tweets in Italy. In total, around 400,000 Italian tweets are contained in the data set, of which around 12,000 have a keyword. Similar to the over-all curves described in section 5.1, the sentiment curve slowly decreases over time, keywords are not used frequently before the end of January, when the first cases in Italy were confirmed. Sentiment in the keyword tweets starts out very negative and then increases again. Interestingly, we see a dip in sentiment on March 9, which is exactly when the Italian lockdown was announced. Keywords were also used most frequently during that week. The dip is not visible in the keyword-only sentiment curve, suggesting that the negative sentiment was actually caused by the higher prevalence of coronavirus-related tweets.

### 5.2.2 Spain

For Spain, around 780,000 tweets were collected in total with around 14,000 keyword tweets. The curves are shown in figure 8. The heavier usage of keywords starts around the same time as in Italy, where the first domestic cases were publicized at the same time. The spike in keyword-only sentiment in mid-February is actually an artifact of the low number of keyworded tweets in combination with the fact that "corona" is a word with other meanings in Spanish (in contrast to the other languages). With more keyword mentions, the sentiment drops as in other countries.

From there onwards, the virus progressed somewhat slower in Spain, which is reflected in the curves as well. A lockdown was announced in Spain on March 14, corresponding to a dip in the sentiment curve. As with the Italian data, this dip is not present in the keyword-only sentiments.

### 5.2.3 France

Analyses for the data from France are shown in figure 9. For France, around 309,000 tweets and around 4,600 keyword tweets were collected. Due to the lower number of data points, the curves are somewhat less smooth. Despite the first European COVID-19 case being detected in France in January, cases did not increase significantly until the end of February, which once again is also seen in the start of increased keyword usage here. The French lockdown was announced on March 16 and extended on April 13, both reflected in dips in the sentiment curve. Towards the end of the considered period, keyword-only sentiment actually starts to increase, which is also seen in Italy and Germany. This could indicate a shift to a more hopeful outlook with regards to the pandemic.

### 5.2.4 Germany

For Germany, around 415,000 tweets and around 5,900 keyword tweets were collected. The analysis results are shown in figure 10. After very few first cases at the end of January, Germany's case count did not increase significantly until early March, which is again when keyword usage increased. The decrease in the sentiment curve actually arrives around the same time as in France and Spain, which is a little surprising because social distancing measures were not introduced by the government until March 22 (extended on March 29). German users were likely influenced by the situation in their neighboring countries here. In

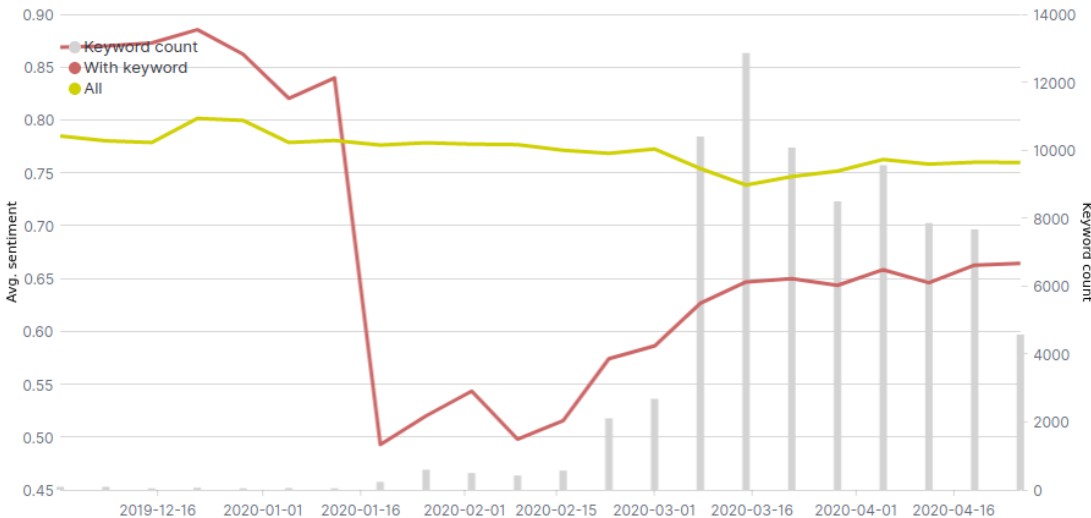

Figure 5: Development of average sentiment for all tweets and for tweets containing COVID-19 keywords, and development of number of tweets containing COVID-19 keywords.

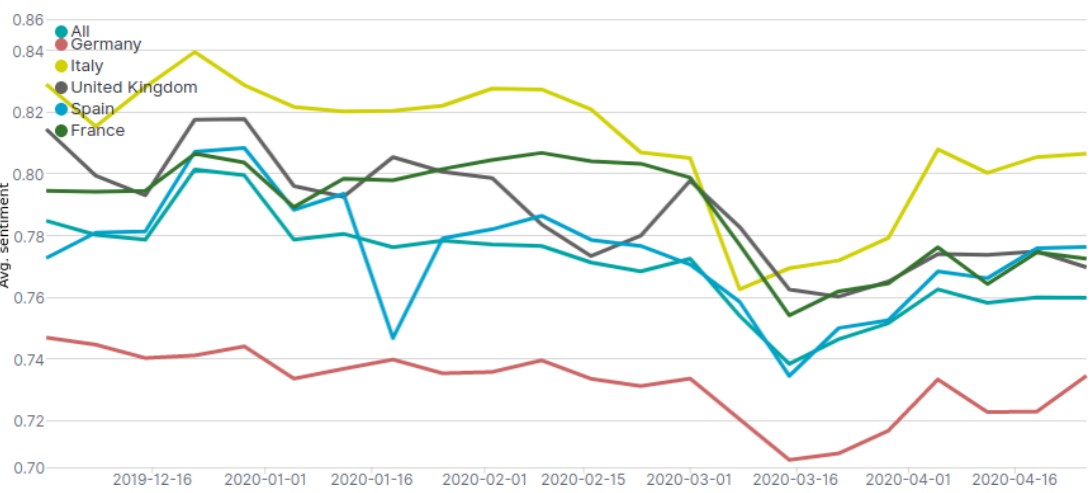

Figure 6: Development of average sentiment over time by country (all tweets).

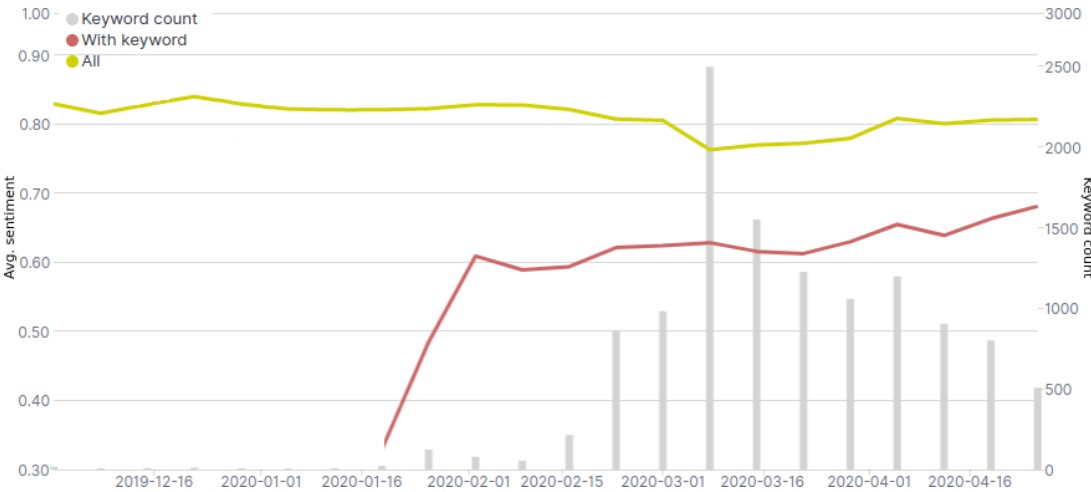

Figure 7: Italy: Development of average sentiment for all tweets and for tweets containing COVID-19 keywords, and development of number of tweets containing COVID-19 keywords.

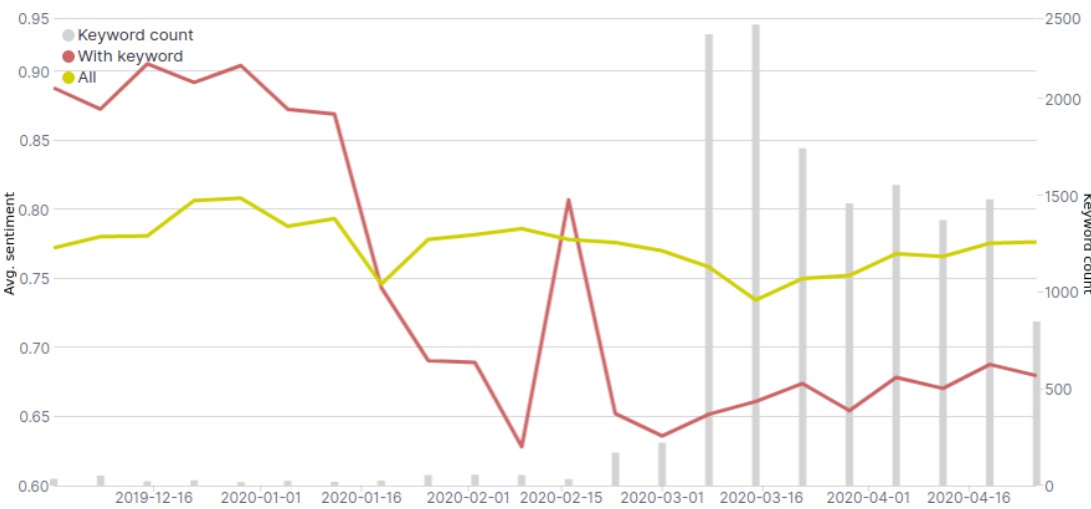

Figure 8: Spain: Development of average sentiment for all tweets and for tweets containing COVID-19 keywords, and development of number of tweets containing COVID-19 keywords.

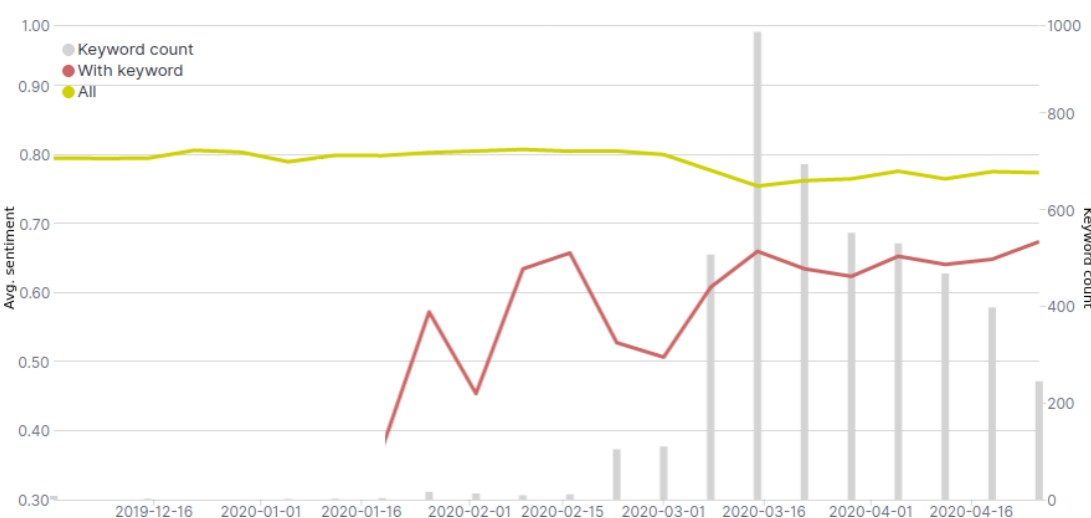

Figure 9: France: Development of average sentiment for all tweets and for tweets containing COVID-19 keywords, and development of number of tweets containing COVID-19 keywords.

general, the curve is flatter than in other countries. One possible reason for this might be the lower severity of measures in Germany, e.g. there were no strict curfews.

In contrast to all other considered countries, the keyword-only sentiment curve is not significantly below the sentiment curve for all tweets in Germany after the beginning of March. There are some possible explanations for this. For one, governmental response to the situation was generally applauded in Germany (Betsch et al., 2020), and, as mentioned above, was not as strict as in other countries, possibly not impacting people as much. On the other hand, the over-all German curve is lower than its counterparts from other countries, i.e. German tweets have lower average sentiment values in general, possibly caused by cultural factors.

### 5.2.5   United Kingdom

Curves for the United Kingdom are shown in figure 11, calculated on around 1,380,000 tweets including around 22,000 keyword tweets. Higher keyword usage starts somewhat earlier here than expected in February, whereas a significant increase in cases did not occur until March. Once again, keyword-only sentiment starts out very negative and then increases over time.

The British government handled the situation somewhat differently. In early March, only recommendations were given, and a lockdown was explicitly avoided to prevent economic consequences. This may be a cause for the sentiment peak seen at this time. However, the curve falls until mid-March, when other European countries did implement lockdowns. The government finally did announce a lockdown starting on March 26. This did not lead to a significant change in average sentiment anymore, but in contrast with other countries, the curve does not swing back to a significantly more positive level in the considered period, and actually decreases towards the end.

### 6   Conclusion

In this paper, we presented the results of a sentiment analysis of 4.6 million geotagged Twitter messages collected during the months of December 2019 through April 2020. This analysis was performed with a neural network trained on an unrelated Twitter sentiment data set. The tweets were then tagged with sentiment on a scale from 0 to 1 using this network. The results were aggregated by country, and averaged over time. Additionally, the sentiments of tweets containing COVID-19-related keywords were aggregated separately.

We find several interesting results in the data. First of all, there is a general downward trend in sentiment in the last few months corresponding to the COVID-19 pandemic, with clear dips at times of lockdown announcements and a slow recovery in the following weeks in most countries. COVID-19 keywords were used rarely before February, and correlate with a rise in cases in each country. The sentiment of keyworded tweets starts out very negative at the beginning of increased keyword usage, and becomes more positive over time. However, it remains significantly below the average sentiment in all countries except Germany. Interestingly, there is a slight upward development in sentiment in most countries towards the end of the considered period.

### 7   Future work

We will continue this study by also analyzing the development in the weeks since May 1st and the coming months. More countries will also be added. It will be very interesting to compare the shown European results to those of countries like China, South Korea, Japan, New Zealand, or even individual US states, which were impacted by the pandemic at different times and in different ways, and where the governmental and societal response was different from that of Europe.

There are also many other interesting research questions that could be answered on a large scale with this data - for example, regarding people's trust in published COVID-19 information, their concrete opinions on containment measures, or their situation during an infection. Other data sets have also been published in the meantime, including ones that contains hundreds of millions of tweets at the time of writing (e.g. Qazi et al., 2020; Banda et al., 2020). These data sets are much larger because collection was not restricted to geotagged tweets. In Qazi et al. (2020), geolocations were instead completed from outside sources.

These studies could also be extended to elucidate more detailed factors in each country. One possibility here is an analysis of Twitter usage and tweet content by country. Another, as mentioned above, lies in moving from the binary sentiment scale to a more complex model.

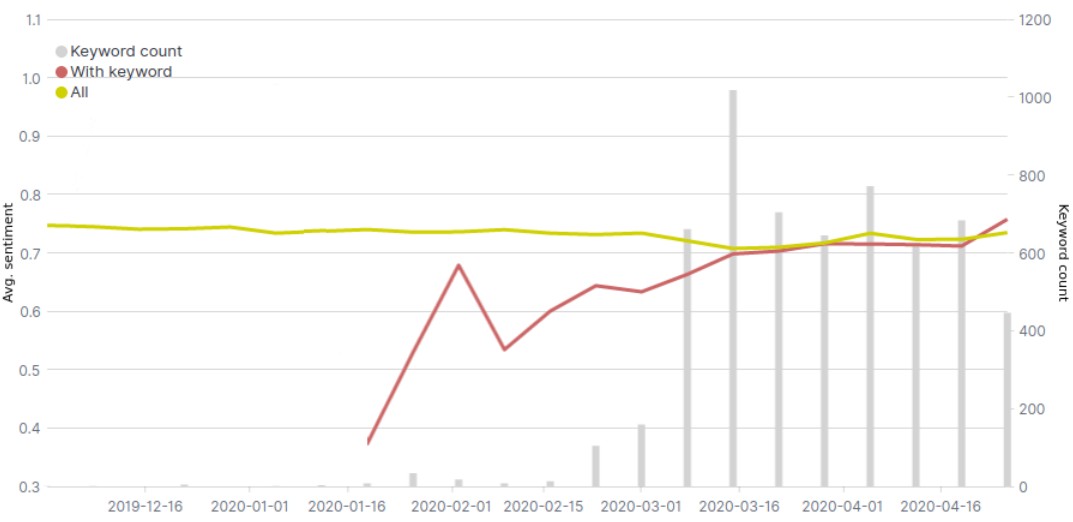

Figure 10: Germany: Development of average sentiment for all tweets and for tweets containing COVID-19 keywords, and development of number of tweets containing COVID-19 keywords.

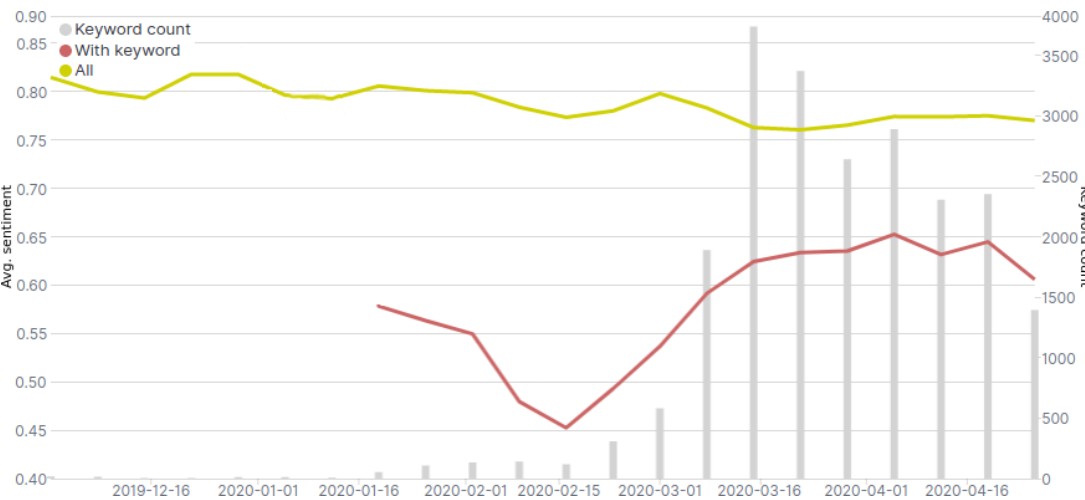

Figure 11: United Kingdom: Development of average sentiment for all tweets and for tweets containing COVID-19 keywords, and development of number of tweets containing COVID-19 keywords.

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
