# OpenReview forum: "Cross-language sentiment analysis of European Twitter messages during the COVID-19 pandemic"
_aclweb.org/ACL/2020/Workshop/NLP-COVID — NLP-COVID-2020_

### Official Review · AnonReviewer3 · 2020-07-04

**Rating:** 6
**Confidence:** 5

**Review:**

This is a mostly well-written overview of an exercise to assign a sentiment label to the European-country generated tweets during the period December’19-May’20.

The authors describe how they differentiate and identify the country, how they assign the sentiment level (positive, neutral, negative), how they use emojis, and how they use the deep learning neural model which presumably can adjust this label assignment regardless of what language the tweet is originally written. The authors report a 0.82 accuracy of their system. The rest of the paper is a recognition of the limitations, and a description and plotting of the sentiment level for various European countries.

Unfortunately, these results do not contribute to adding new knowledge.  The study could use more work.

Suggestions:

Could the authors provide a breakdown by language of the tweets that they process? Are we to assume that all tweet originated from Italy are in Italian and those originating in Germany are in German?

Is this data publicly available?

Has the 0.82 accuracy been manually validated? Is there a difference in accuracy depending on the language? The authors claim that one of the contributions of their study is this tagged dataset (geotagged, and sentiment-tagged). It seems there is no further evaluation on how well the tagging has been applied.

And while it is visibly clear that we see a global fall in sentiment that correlates with governments issuing lock-down protective measures, and this result could be a start that this labelling of the data is good, is there anything else we can say, is there any other way we can analyze this data and identify common topics in the similar sentiment groups? Something that can be actually useful to the COVID-19 researchers…

---

### Official Review · AnonReviewer1 · 2020-07-04
**Review on "Cross-language sentiment analysis of European Twitter messages during the COVID-19 pandemic"**

**Rating:** 6
**Confidence:** 5

**Review:**

The authors carried on a deep learning pipeline to analyze the sentiment of Twitter texts, and propose a complete research. The presentation and language part of this submission is good.

 However, the research mainly use the routine DL methodology and the analysis method is not contributive.  In general, the novelty and contribution of this research do not reach the level of publication as a ACL workshop paper. Here comes some comments and suggestions.

1. The data statistics is missing. Though we found a rough number list in Figure 1, they are not quite clear. Data with time series info are also welcomed. Furthermore, several python packages help to draw Europe Map, and might make this part more vivid.
2. It is better to provide a figure to explain the structure of the network. The authors surely already gave some details in page 2, including the input layer, activation function info. The hyper parameter of the network could also be provided.
3. It is lacking of comparison of the current NN with some other NN structure. How would one single experiment derive convincing result without baseline methods or intrinsic evaluation? This is a core question I would like to raise here for this research.
4. I am thinking of a possibility of splitting the Twitter data in terms of weeks, and take time series consideration into the current research paradigm. A sentiment-time curve plot might lead to some instructive hypothesis, if the research take a more sophisticated experiment design.

---

### Official Review · AnonReviewer2 · 2020-07-06
**Review of "Cross-language sentiment analysis of European Twitter messages"  -- interesting trends analysis but some more approach comparisons and tables for the data would be good.**

**Rating:** 6
**Confidence:** 3

**Review:**

The authors present an interesting, important and relevant trend analysis of sentiment across languages in several locales during the Covid-19 pandemic, using geo-tagged European Twitter data and pre-trained cross-lingual embeddings within a neural model.

The main contributions of the paper are: 1) the geo-tagged European Twitter dataset of 4.6 million tweets between Dec 2019 and Apr 2020, where some of these contain Covid19-specific keywords (it would be nice to see some percentage breakdown stats by language here), and 2) the important trends by country in terms of dip and recovery of sentiment over this period, including the overall trends across the board.

In terms of sentiment modeling, they use a pre-trained neural model trained on the Sentiment140 dataset of Go et al, which is English-only, hence they freeze the weights to prevent over-adapting to English. They use cross-lingual MUSE embeddings to train this network to better generalize sentiment prediction to multi-lingual data for each country. There is no novelty in the modeling approach itself, which works for the purposes of trend analysis being performed. However, there is no comparison being presented of results of experimentation with different approaches, to corroborate or contrast their current trends results. E.g. a simple baseline approach could have been to run Average and Polarity sentiment values using a standard python text processing package such as `textblob` to obtain sentiment predictions. Other experiments could have been done to use different pre-trained embeddings such regular GloVE or Multi-lingual BERT to provide a comparison or take the average of the approaches to get a more generalized picture of sentiment trends. Also the authors should make it clear that the model has really been used in perhaps inference mode only to obtain the final sentiment predictions for each tweet.

The treemap visualization gives a good overall picture of tweet stats, but a table providing the individual dataset statistics including keywords chosen by locale would be really helpful.

Some notable trends are how the sentiment generally dips in all locales right around the time of lockdown announcements, and recovers relatively soon after, except for Germany where it dips at the same time as neighboring countries despite lockdown being started here much later, and UK, where sentiment stays low. It is also interesting to note the spikes and fluctuations in Covid19-related sentiment for Spain, and the overall trend for average sentiment by country for "all" tweets (including Covid19-related ones) tracking similarly over the time period considered.

However, one trend it would be good to see some discussion on is how the histogram of keywords correlate with the sentiment for the keyworded tweets, as it appears interesting that heightened use of Covid-19 keywords in tweets tracks with more positive sentiment in most of the plots. Perhaps it would be helpful to have a separate discussion section for the overall trend analysis at the end.

Overall the paper is well-motivated and in its current form provides perhaps the intended insights, and presents lot of scope to perform useful extended analyses with more meaningful comparisons for additional time spans and across countries where governmental and societal response were different than in Europe. Perhaps the authors could consider a more interpretable predictive sentiment model in future with some hand-crafted features such as geotag metadata, unigram and bi-gram features, binary features for government measures, and Covid19-specific keyword features by locale, which could provide more insight into why sentiment predictions trend a certain way during a specific period for a given locale.

---

### Comment · Program_Chairs · 2020-07-06
**Revise and Resubmit + presentation**

The authors have trained a sentiment model which is then applied to a large data set of COVID-19-related tweets to characterize sentiment trends in different European countries.

The analysis is interesting, but the inferences are largely circumstantial. The "correlation" presented is not a statistical analysis but rather a rough association with public events; stronger discussion of the limitations of using Twitter data to make inferences about population-level sentiment is needed.

Overall, the work is interesting but for a long paper submission requires a bit more analytical or methodological rigour. I think this could be added in a revision.

At this stage, I am recommending that the authors present the work at the workshop on Thursday (please prepare a 10-minute presentation), but cannot finalise the acceptance decision.

---

### Decision · Program_Chairs · 2020-10-15

**Decision:**

Accept

**Comment:**

Entering decision previously communicated after revision.